# The Toxicity of Mancozeb Used in Viticulture in Southern Brazil: A Cross-Sectional Study

**DOI:** 10.3390/ijerph23010034

**Published:** 2025-12-25

**Authors:** Sheila de Castro Cardoso Toniasso, Camila Pereira Baldin, Vittoria Calvi Sampaio, Raquel Boff da Costa, Nelson David Suarez Uribe, Patrícia Gabriela Riedel, Débora Costa, Norma Marroni, Elizângela Schemitt, Marilda Brasil, Ana Leticia Hilário Garcia, Juliana da Silva, Eliane Dallegrave, Maria Carlota Borba Brum, Robson Martins Pereira, Franciele Lopes dos Reis, Luciana da Silva Pereira, Eduardo Natan Maraschin Klein, Hidayat Kassim, Dvora Joveleviths

**Affiliations:** 1Graduate Program in Gastroenterology and Hepatology, Federal University of Rio Grande do Sul, Porto Alegre 90035-003, RS, Brazil; cbaldin@hcpa.edu.br (C.P.B.); visampaio@hcpa.edu.br (V.C.S.); raquelboff80@gmail.com (R.B.d.C.); nduribe@hcpa.edu.br (N.D.S.U.); priedel@hcpa.edu.br (P.G.R.); deboranutri.diehm@gmail.com (D.C.); mcbrum@hcpa.edu.br (M.C.B.B.); rompereira@hcpa.edu.br (R.M.P.); flreis@hcpa.edu.br (F.L.d.R.); lpesilva@hcpa.edu.br (L.d.S.P.); edukak@gmail.com (E.N.M.K.); hidayatkassim@hotmail.com (H.K.); dvorajov@gmail.com (D.J.); 2Experimental Laboratory of Pulmonary Science and Inflammation, Hospital de Clínicas de Porto Alegre, Porto Alegre 90035-003, RS, Brazil; npmarroni@hcpa.edu.br (N.M.); eschemitt@hcpa.edu.br (E.S.); msbrasil@hcpa.edu.br (M.B.); 3Genetic Toxicology Laboratory, La Salle University (UniLaSalle), Canoas 92425-900, RS, Brazil; ana.garcia@unilasalle.edu.br (A.L.H.G.); juliana.silva@unilasalle.edu.br (J.d.S.); 4Department of Pharmacosciences, Federal University of Health Sciences of Porto Alegre, Porto Alegre 90050-170, RS, Brazil; elianedal@ufcspa.edu.br

**Keywords:** biomonitoring, ETU biomarker, genotoxicity, mancozeb, occupational health, oxidative stress, pesticide exposure, vineyard workers

## Abstract

Background: Viticulture in Southern Brazil heavily relies on fungicides, such as Mancozeb, to manage fungal diseases. Increasing concern has emerged regarding the chronic health effects of Mancozeb exposure among vineyard workers, particularly its potential to induce oxidative stress and genotoxicity. Methods: A cross-sectional study was conducted between July and November 2023 involving 94 participants: 50 vineyard workers occupationally exposed to Mancozeb and 44 organic farmers with no history of pesticide exposure, who served as the control group. Eligible participants were aged 18 years or older, and exposed individuals had at least 5 years of documented Mancozeb use. Data on demographics, health status, occupational history, and use of personal protective equipment (PPE) were collected through structured interviews. Blood and urine samples were analyzed to determine hematological and biochemical parameters, oxidative stress biomarkers, genotoxicity (via comet assay and micronucleus test), and urinary ethylene thiourea (ETU), the primary metabolite of Mancozeb. Results: Workers exposed to Mancozeb exhibited significantly elevated levels of oxidative stress markers (*p* < 0.001) and DNA damage in both genotoxicity assays (*p* < 0.001). Urinary ETU concentrations were also markedly elevated, and a threshold of 69.3 ng/mL was identified as a discriminative marker of exposure. Conclusions: This study offers a novel contribution by proposing a specific biological exposure limit for ETU concentrations, derived from ROC curve analysis, representing a significant advancement in occupational health. The findings underscore the urgent need for regulatory biological exposure limits and the implementation of effective preventive strategies.

## 1. Introduction

Viticulture has undergone accelerated growth over the years. Southern Brazil is the country’s leading region for grape cultivation and winemaking, with the Serra Gaúcha accounting for over 70% of the cultivated area [1]. Brazilian viticulture originated in this southern region, where it evolved into one of the most significant economic and cultural activities. This tradition was introduced by Italian immigrants who arrived in Brazil in the late 19th century, bringing with them technical expertise in grape and wine production, along with a profound cultural connection to the land and vine cultivation.

These families of Italian descent settled on small rural properties and implemented production systems based on family labor, preserving agricultural practices and lifestyles passed down through generations. Over time, viticulture became a central element of the economies and daily lives of many communities. Thus, it transcends its productive role and embodies the cultural identity of Italian descendants, enriching Brazil’s cultural diversity.

To sustain large-scale production, this agricultural system relies heavily on chemical inputs such as fertilizers and pesticides. Mancozeb is a fungicide and acaricide commonly used on grapevines. It ranks fourth among the most widely used pesticides in Brazil [2,3]. Although it is widely recognized that Mancozeb does not significantly bioaccumulate in soil or water, inadequate regulation of its sale and use, combined with the lack of personal protective equipment (PPE), can lead to various health and environmental issues, particularly for agricultural workers exposed to this compound [4].

Several biomarkers are essential tools for monitoring the physiological and biochemical impacts of toxic substances. The metabolic degradation of ethylenebisdithiocarbamates (EBDCs), including Mancozeb, results in the formation of carbon disulfide (CS_2_), ethylene thiourea (ETU), manganese (Mn), and zinc (Zn). ETU is considered a key biomarker for assessing exposure to this pesticide. Another important marker is acetylcholinesterase, an enzyme responsible for regulating nerve impulses by breaking down acetylcholine at neuromuscular junctions and synapses, which may be altered during acute pesticide poisoning [5,6,7]. Although acetylcholinesterase (AChE) is traditionally used as a biomarker for organophosphate exposure, dithiocarbamates can also affect its enzymatic activity. Estimates of ETU elimination kinetics vary significantly, ranging from 32 to 100 h [8,9].

The toxicity of these compounds can also be evaluated through oxidative stress, characterized by an imbalance between excessive production of reactive oxygen species (ROS) and cellular antioxidant defense systems [10]. Scientific evidence suggests that prolonged exposure to pesticides may be linked to DNA damage and cancer development. Studies have demonstrated that pesticides can induce genotoxicity, impair DNA repair mechanisms, and increase the risk of persistent mutations [11].

Exposure to the fungicide Mancozeb has been linked to pronounced hepatotoxic effects, including oxidative stress, disruptions in essential metal homeostasis, and histopathological injury in rodent experimental models [12,13]. Evidence indicates that the compound triggers biochemical and inflammatory disturbances in hepatic tissue, underscoring its toxic potential even under subchronic exposure conditions [14].

Although its carcinogenic potential remains under debate, Mancozeb is believed to contribute to chronic liver damage, potentially affecting hepatic metabolism and gene expression involved in detoxification pathways [15]. These findings highlight the importance of biochemical and clinical monitoring of workers chronically exposed to pesticides, especially in regions with intensive agricultural activity.

Accordingly, this study was developed as part of a broader translational research project, following a previous experimental study conducted by the same research group. It aims to evaluate the toxicity of the fungicide Mancozeb, an ethylenebisdithiocarbamate (EBDC), in agricultural workers exposed and unexposed to this substance in Southern Brazil.

## 2. Materials and Methods

This cross-sectional study was conducted between July and November 2023, coinciding with the period of grapevine spraying with Mancozeb. A total of 94 participants were recruited: 50 grape producers from Southern Brazil were included in the exposed group, and 44 organic farmers from the same region were included in the unexposed group.

Inclusion criteria for the exposed group were as follows: age ≥ 18 years and occupational use of Mancozeb for at least five years. For the unexposed group, the following criteria were used: age ≥ 18 years, working in certified organic farming for at least five years, self-declared organic farmers, and holding certification or a conformity assessment accredited by regulatory agencies. Workers with active neoplastic diseases were excluded.

This study was approved by the Research Ethics Committee of the Hospital de Clínicas de Porto Alegre, Brazil (CAAE No. 11627319.5.0000.5327), and all participants provided written informed consent before enrollment.

### 2.1. Data Collection

Data was collected using a structured questionnaire addressing sociodemographic information, educational level, work practices, use of personal protective equipment (PPE), duration of exposure to Mancozeb, health status, and current medications. The Alcohol Use Disorders Identification Test (AUDIT) was applied to assess alcohol consumption. Anthropometric data and physical examinations were also performed on all participants. Additionally, rapid tests for hepatitis B and C were administered. The farmers wore long-sleeved shirts, gloves, long pants, and masks. Mancozeb was applied every 15 days from September to December.

Blood samples were collected to assess hematological parameters, genotoxicity, and oxidative stress. Samples were collected in the field, centrifuged, separated, stored on ice, and transported for subsequent analysis to Porto Alegre.

### 2.2. Biochemical and Hematological Analyses

The following biochemical parameters were assessed: urea and creatinine (colorimetric method); aspartate aminotransferase (AST) and alanine aminotransferase (ALT) (enzymatic method); alkaline phosphatase (kinetic colorimetric method—p-NNP—DG KC); complete blood count and platelet count using light absorption spectrophotometry, impedance, and flow cytometry; and BChE activity (kinetic enzymatic method).

### 2.3. Oxidative Stress

Lipid peroxidation levels were measured in serum using the TBARS assay and expressed in nmol/mg protein, as described by Buege and Aust [16]. Protein concentration was determined using the Bradford method [17]. The activities of antioxidant enzymes SOD and CAT were measured according to the protocols of Misra and Fridovich [18] and Boveris and Chance [19], with results expressed as specific enzymatic activity units.

### 2.4. Genotoxicity Comet Assay

The alkaline comet assay was performed according to the protocol described by Souza et al. [20], with minor modifications. Blood samples (5 µL) were gently mixed with 95 µL of 0.75% low-melting-point agarose and immediately spread on microscope slides previously coated with 1.5% normal-melting-point agarose (Appendix A) [21].

### 2.5. Micronucleus Test in Exfoliated Buccal Mucosa Cells

Buccal cells were collected using cervical cytobrushes by scraping the right and left sides of the oral mucosa of each participant.

The micronucleus test followed the protocol outlined by Thomas et al. [22]. After cell collection, the brushes were immersed in conical tubes containing 10 mL of Saccomanno’s fixative solution (Appendix A). The slides were subsequently stained with Feulgen and Light green, and 2000 cells per participant were microscopically analyzed. The analysis considered the frequency of micronuclei (MN), nuclear buds (BUD), broken eggs (BE), and binucleated cells.

### 2.6. Urine Analysis and ETU Detection

Results were expressed in nanograms (ng) of ETU per milliliter (mL) of urine.

ETU levels in urine were determined using liquid chromatography coupled with tandem mass spectrometry (UHPLC-MS/MS). Urine samples were stored in polypropylene tubes at −80 °C until analysis. The method used was a modified QuEChERS protocol, followed by UHPLC-MS/MS analysis [23] (Appendix A).

### 2.7. Statistical Analysis

A sample size of 86 participants (43 per group) was calculated to detect a minimum mean difference of 0.7 units between exposed and unexposed groups, assuming a standard deviation of 1 unit, 90% power, and a 5% significance level. An additional 10 participants were included to account for potential dropouts, bringing the total to 96.

Quantitative variables were described using means and standard deviations or medians and interquartile ranges, as appropriate. Categorical variables were described using absolute and relative frequencies.

Student’s *t*-test was used for mean comparisons. In cases of asymmetry, the Mann–Whitney U test was applied. Pearson’s chi-square or Fisher’s exact test was used for comparing proportions.

Receiver Operating Characteristic (ROC) curve analysis was used to determine the optimal cut-off point for urinary ETU in identifying pesticide exposure.

To control potential confounders, multivariate linear regression models were applied. Regression coefficients (β) and their 95% confidence intervals (CIs) were calculated to estimate the effect of exposure on the studied outcomes. Variables with a *p*-value < 0.20 in the bivariate analysis and with practical relevance in the literature were included in the multivariate models.

All raw data used in the statistical analyses are available from the corresponding author, ensuring transparency and enabling future systematic reviews and meta-analyses.

The significance level was set at 5% (*p* < 0.05), and analyses were performed using SPSS version 27.0.

## 3. Results

A total of 94 agricultural workers were analyzed, comprising 50 in the exposed group and 44 in the non-exposed group (Table 1).

In the exposed group, the majority of participants were male (92%). The two groups were similar in terms of mean age, ethnicity, educational level, and smoking status.

Although neither group engaged in regular physical activity at levels considered significant (i.e., more than 150 min per week), both were classified as physically active due to the nature of their work, which involves physical exertion and frequent walking.

The exposed group had a higher prevalence of hypertension, dyslipidemia, and use of medications. In contrast, musculoskeletal disorders were more common among participants in the unexposed group.

All workers in the exposed group reported using personal protective equipment (PPE), including gloves, long pants, masks, boots, and long-sleeved shirts.

Among the non-exposed group, predominant crops included vegetables and leafy greens.

The mean duration of pesticide exposure in the exposed group was 28 years, ranging from 19.8 to 40 years.

Approximately 50% of participants in the unexposed group reported prior exposure to pesticides, with their last exposure occurring more than five years earlier.

Moreover, 86% of workers in the exposed group used backpack sprayers to apply Mancozeb during vineyard activities.

Low-dose alcohol consumption was more prevalent in the exposed group.

There were no significant differences between the two groups in analyses of urea, creatinine, aspartate aminotransferase (AST), alanine aminotransferase (ALT), and alkaline phosphatase, all within normal limits.

Participants in the exposed group had a larger abdominal circumference, indicating greater central fat accumulation, and consistently higher blood pressure—both systolic and diastolic—possibly suggesting increased cardiovascular risk. Other vital signs and Body Mass Index (BMI) did not show meaningful differences between the groups.

The results presented in Table 2 show clear biological effects associated with occupational exposure to Mancozeb among vineyard workers.

Urinary ETU levels were significantly higher in the exposed group than in the unexposed group (*p* < 0.010), confirming effective internal exposure and validating ETU as a reliable biomarker for assessing Mancozeb exposure.

Biochemical parameters, ◦, in Figure 1, indicated a pronounced oxidative imbalance in the exposed group. Increased TBARS levels (*p* < 0.001) and catalase activity (*p* < 0.001), coupled with a significant reduction in SOD activity (*p* < 0.001), suggest enhanced lipid peroxidation and an altered antioxidant response, consistent with oxidative stress induced by pesticide exposure. Protein, *p* = 000.1.

Genotoxic biomarkers also showed notable differences. Although the frequencies of binucleated cells and micronuclei did not reach statistical significance between groups, the DNA damage index (assessed by the comet assay; visual score; Figure 2) was significantly higher in the exposed group (*p* < 0.001), supporting the genotoxic potential of Mancozeb in real-world occupational conditions.

The comet assay results (Figure 2) showed significantly greater DNA damage in the Mancozeb-exposed group than in the unexposed group (*p* < 0.001). This finding is consistent with previous studies demonstrating Mancozeb’s genotoxic potential in both in vitro and in vivo models. The elevated DNA damage index reinforces the evidence of genotoxic stress among occupationally exposed individuals, highlighting the need for preventive measures and regular biological monitoring in this worker population.

Furthermore, although BUD + BE and micronuclei frequency did not differ significantly, there was a tendency toward greater chromosomal instability in the exposed group (Figure 3).

Overall, the findings indicate that chronic occupational exposure to Mancozeb is associated with increased systemic oxidative stress and DNA damage in exposed workers, emphasizing the need to improve protective measures and implement biological monitoring protocols in agricultural environments.

Micronucleus frequency is a widely used biomarker for indicating genetic damage or genomic instability.

The exposed group showed a significantly higher mean number of MN/2000 cells than the unexposed group.

This result suggests that individuals in the exposed group (likely due to pesticide exposure, as described previously) exhibited greater genotoxic damage, potentially as a consequence of chronic exposure to pesticides such as Mancozeb.

The increase in MN levels in the exposed group may be related to prolonged exposure (a mean of 28 years) and use of spraying equipment, despite PPE use.

These findings underscore the importance of biological monitoring and preventive measures to reduce occupational genetic risk in this population.

Table 3 presents the results of a multivariate linear regression analysis evaluating the effect of occupational exposure on various biochemical and biological outcomes. The model was adjusted for potential confounding variables, such as hypertension, dyslipidemia, alcohol use, elevated BMI, musculoskeletal disorders, age, and sex (as indicated in the table footnotes).

The multivariate analysis did not alter the previously observed statistically significant associations between occupational Mancozeb exposure and oxidative stress and genotoxic damage.

Although transaminase levels did not differ significantly, the possibility of Mancozeb-related hepatotoxicity cannot be ruled out.

Exposure was significantly associated with increased urinary ETU, a specific biomarker of Mancozeb exposure.

Elevated TBARS levels indicated enhanced lipid peroxidation and oxidative stress. The significant increase in catalase activity, coupled with the marked reduction in SOD, indicates an imbalance in the antioxidant defense system.

The variable “damage,” associated with the presence of micronuclei or other genotoxic markers, was significantly higher in the exposed group, consistent with the previous figures.

Although no significant differences were observed in classical hepatic enzymes, such as AST and ALT, this finding warrants further discussion. The absence of detectable alterations in these markers may indicate that hepatic damage associated with Mancozeb exposure occurs subclinically, without evident biochemical manifestation, or that it is expressed through mechanisms not captured by conventional enzymes, such as mitochondrial dysfunction or subcellular inflammation mediated by oxidative stress. Furthermore, given that the analyzed cohort was predominantly male, it is important to acknowledge that potential gender-related differences in xenobiotic metabolism and pesticide susceptibility may influence hepatic responses. Exploring these differences could contribute to a broader understanding of Mancozeb’s impact on public health, emphasizing the need for future studies with more gender-balanced samples.

ALT, creatinine, total cholesterol, and LDL were not significantly associated with exposure.

The Receiver Operating Characteristic (ROC) curve assesses the accuracy of ETU as a biomarker of Mancozeb exposure. The area under the curve (AUC) was 0.95 (95% CI: 0.90–1.00), indicating excellent discriminative ability to distinguish between exposed and non-exposed individuals.

The identified cut-off value was 69.3 ng/mL, with 100% sensitivity and 90.2% specificity. These values demonstrate that ETU accurately identified all exposed individuals (no false negatives) and produced a low false-positive rate among unexposed individuals.

In Figure 4, the cut-off value of 69.3 ng/mL was determined via comparison with the control group of organic farmers with no occupational exposure to pesticides and reflects the upper limit observed in non-exposed individuals from the same region.

The high sensitivity and specificity suggest that ETU is a valuable biomarker for the occupational screening and biological monitoring of workers exposed to Mancozeb, despite the limitation that no universally recognized specific marker currently exists.

Occupational exposure to pesticides, particularly Mancozeb, was associated with increased oxidative stress—evidenced by elevated TBARS and altered antioxidant enzyme activity (increased catalase and decreased SOD)—greater genotoxic damage (increased DNA damage markers),and confirmed internal exposure via elevated urinary ETU.

## 4. Discussion

This innovative study assessed the human toxicity of the pesticide Mancozeb and forms part of a broader translational research initiative. Previous experimental data in rats demonstrated the compound’s toxic potential, particularly its direct effects on gut microbiota [24].

A 2021 literature review linked Mancozeb to various adverse health outcomes, including hepatic, renal, and genotoxic effects, with elevated levels of ETU and liver enzymes [25]. These findings align with the results of the present study, which identified alterations in antioxidant systems associated with oxidative stress and genotoxic damage, as well as increased urinary ETU concentrations.

Grapevine cultivation under conditions of high humidity and mild temperatures—characteristic of Southern Brazil’s climate—promotes fungal proliferation and necessitates preventive disease control. In such scenarios, chemical control with fungicides such as Mancozeb is commonly adopted [26].

The detection of Mancozeb as a food residue raises additional concerns regarding exposure, as it may contribute to ETU levels in individuals considered non-exposed.

A study on glyphosate exposure revealed that workers who did not wear gloves exhibited higher urinary glyphosate levels than those who did. Moreover, manual application was significantly associated with elevated biomarker concentrations [27].

Mancozeb toxicity is primarily attributed to its metabolite, ethylene thiourea (ETU). Prior studies have demonstrated a correlation between urinary ETU levels and pesticide application. The use of spraying equipment also appears to increase exposure risk. In this study, 86% of workers exposed to Mancozeb reported using sprayers during application.

Additionally, the literature on other pesticides, such as glyphosate, has documented DNA and chromosomal damage in human cells, along with genotoxic, hormonal, and enzymatic effects [28,29]. A 2021 study proposed a potential link between glyphosate exposure and carcinogenesis, identifying it as a possible contributor to non-Hodgkin lymphoma in exposed individuals [30].

Mental health disorders, including depression, have also been associated with pesticide exposure. A U.S.-based study found that farmers exhibited a higher prevalence of severe depression compared to other occupational groups and reported a consistent association between pesticide exposure and depressive symptoms across most studies analyzed [31].

In 2019, the World Health Organization recognized pesticide exposure as a significant risk factor for suicide in rural communities.

Two experimental studies in mice indicated that physical exercise may enhance antioxidant mechanisms, suggesting a potential protective effect in populations exposed to pesticides. In occupational settings, regular physical activity could mitigate the harmful metabolic effects of these substances [32,33].

An experimental study demonstrated that Mancozeb increases oxidative stress by altering antioxidant enzyme levels and elevating hepatic enzyme activity, including aspartate aminotransferase (AST), alanine aminotransferase (ALT), and alkaline phosphatase [34].

Pesticide exposure has also been associated with a 30% increased risk of metabolic syndrome, according to a systematic review encompassing 12 studies and 6789 participants [35].

Although this study focused on Mancozeb, it is worth noting that other widely used agricultural pesticides also exhibit genotoxic and oxidative properties. However, a detailed comparison across pesticide classes lies beyond the scope of this work.

Organophosphate poisoning among agricultural workers has been linked to reduced acetylcholinesterase activity and increased activities of SOD, CAT, and GPx [31]. A longitudinal study reported elevated liver function markers in agricultural workers, with a statistically significant correlation between prolonged pesticide exposure and increased biomarkers of hepatic damage [36].

Several studies have suggested a potential association between chronic pesticide exposure and an increased risk of neurodegenerative diseases such as Parkinson’s and Alzheimer’s. Neurotoxic agents—including organophosphates, carbamates, pyrethroids, and dithiocarbamates—can cross the blood–brain barrier and induce oxidative stress, chronic inflammation, and mitochondrial dysfunction, all of which are implicated in the pathophysiology of these conditions [37,38].

In Parkinson’s disease, occupational pesticide exposure is linked to the selective degeneration of dopaminergic neurons in the substantia nigra. In Alzheimer’s disease, evidence from animal models suggests accelerated accumulation of β-amyloid plaques and tau protein tangles following exposure to neurotoxic compounds [39,40].

Furthermore, rural populations or individuals environmentally exposed to pesticides exhibit a higher incidence of early cognitive symptoms, motor deficits, and behavioral disorders, which may represent initial signs of neurodegenerative diseases [41,42]. Long-term exposure to low levels of pesticides, even in the absence of acute poisoning, may result in cumulative effects on the central nervous system [43]. These findings underscore the need for public policies to reduce pesticide use, monitor the health of rural workers, and promote less toxic agricultural alternatives to safeguard neurological health.

The association between noncommunicable chronic diseases (NCDs) and pesticide use has become an increasing concern in public health. Scientific evidence suggests that prolonged and continuous exposure to these chemical agents—commonly used in intensive agriculture—may contribute to the development of various NCDs, including cancers; neurodegenerative diseases (such as Parkinson’s and Alzheimer’s); and endocrine, respiratory, and cardiovascular disorders [44].

DNA damage in exposed workers was identified using both the comet assay and the micronucleus test.

The methodology employed to assess Mancozeb exposure—based on urinary ETU quantification—proved effective in occupational settings. In this context, the ROC curve analysis of urinary ETU presented in this study is particularly noteworthy, as it identifies a specific cut-off point associated with occupational Mancozeb exposure, potentially serving as an estimate of individual toxic burden.

The sample was predominantly male. However, this does not constitute a limitation in evaluating sex-related differences, as the current literature does not indicate significant sex-based variation in the parameters analyzed in relation to Mancozeb exposure.

Future studies by the research group plan to incorporate environmental and dietary assessments and to enhance hepatic toxicity evaluation using FibroScan and non-invasive biomarkers, such as the Fibrosis-4 Index (FIB-4), to detect liver fibrosis.

Over the years, public policies focused on workers’ health have sought to integrate surveillance and prevention strategies addressing the adverse effects of pesticide exposure, particularly among rural workers directly exposed to these substances. It is essential to foster coordination among health services, sanitary and environmental surveillance, and intersectoral initiatives to mitigate the impacts of pesticide use on human health.

Despite regulatory advancements, implementing these policies continues to face several challenges, including underreporting of poisoning cases, deficiencies in monitoring pesticide use and commercialization, and inadequate training of healthcare professionals to identify and report related health outcomes. In this context, it is imperative to strengthen occupational health surveillance systems and promote more sustainable agricultural practices.

The previous literature has already demonstrated the potential toxic risk of Mancozeb, showing a direct relationship between urinary ETU levels and internal dose [45]. Another study reported significant alterations in serum markers of oxidative stress and immune response among vineyard workers exposed to the fungicide, indicating measurable systemic effects [46]. Taken together, these findings underscore the relevance of biomarkers as essential tools for assessing occupational risks and monitoring exposure to Mancozeb.

## 5. Conclusions

Mancozeb is a widely used fungicide in agriculture, and occupational exposure may pose significant health risks to workers. This study identified adverse effects associated with oxidative stress and DNA damage, as well as elevated urinary ETU levels in exposed individuals. Using ROC curve analysis, a biological exposure limit for Mancozeb was established, representing a notable methodological advancement.

The proposed cut-off value for ETU, 69.3 ng/mL, should be considered a central recommendation for biomonitoring programs, serving as a reference parameter for assessing occupational exposure.

The findings reinforce the need for preventive measures, such as substituting less toxic substances, worker rotation, and strict adherence to PPE. This study also highlights the importance of continuous occupational health surveillance amid evolving production processes and the introduction of new pesticides.

## Figures and Tables

**Figure 1 ijerph-23-00034-f001:**
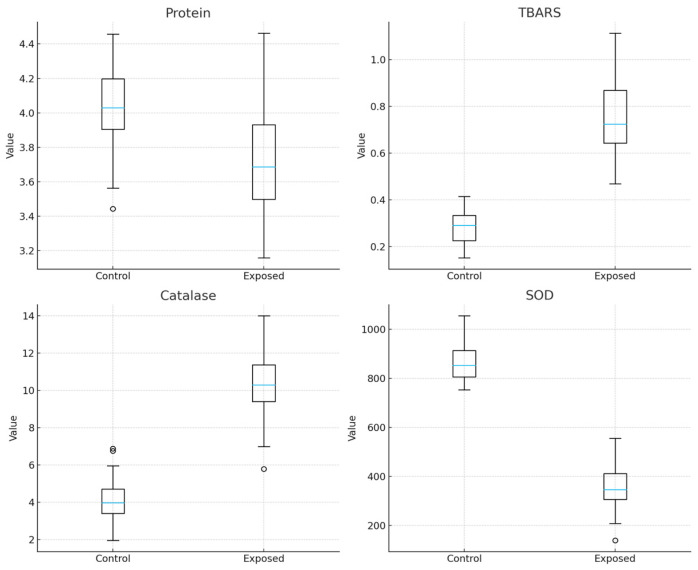
Biochemical parameters indicated a pronounced oxidative imbalance (*p* < 0.001).

**Figure 2 ijerph-23-00034-f002:**
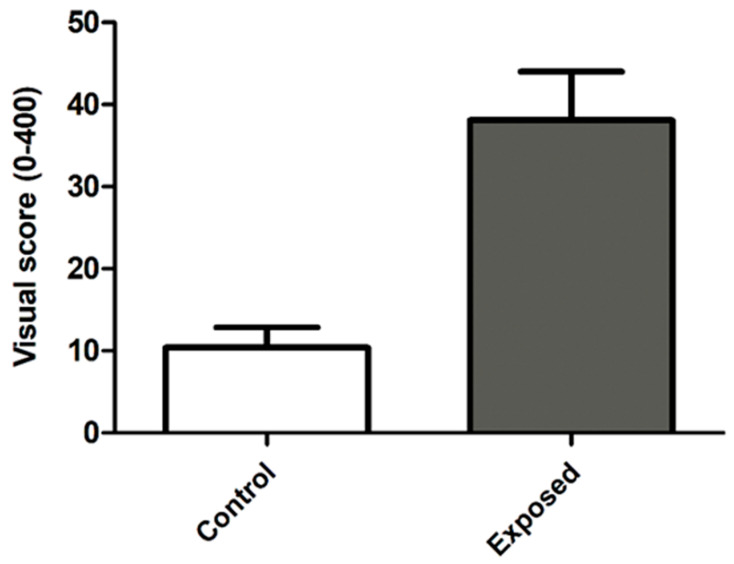
The mean ± standard deviation of the visual score (0–400) assessed using the comet assay in cells from the exposed group compared with the unexposed group. Significant at *p* < 0.001; unpaired *t*-test.

**Figure 3 ijerph-23-00034-f003:**
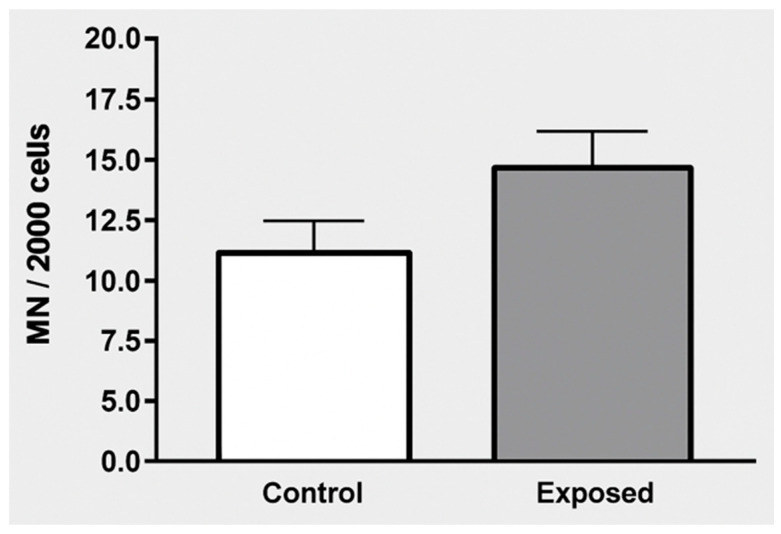
Micronucleus (MN) test results in buccal mucosa cells: unexposed and exposed groups. Data is presented as mean ± standard error per 2000 cells per individual.

**Figure 4 ijerph-23-00034-f004:**
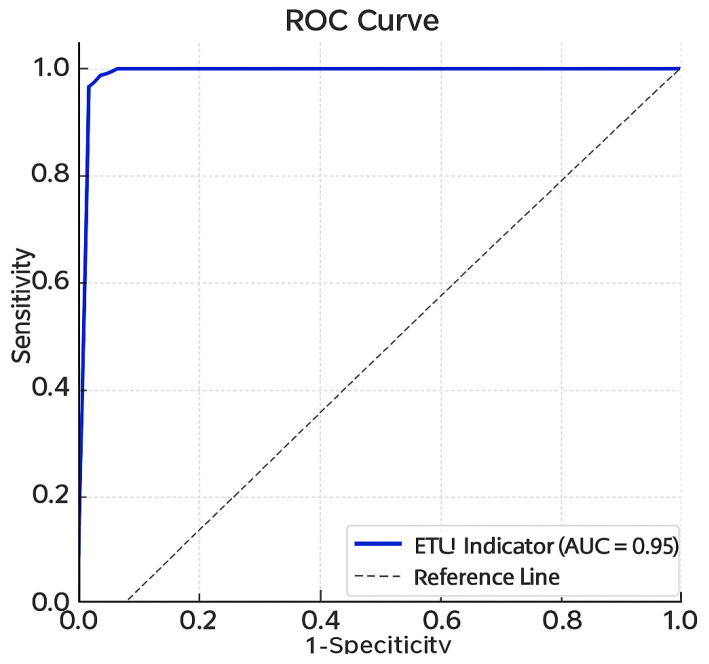
ROC curve of urinary ETU.

**Table 1 ijerph-23-00034-t001:** The sociodemographic, clinical, and lifestyle characteristics of the non-exposed and exposed groups.

Variables	Non-Exposed Group *(n* = 44)	Exposed Group (*n* = 50)	*p*-Value
Age (years)—mean ± SD	49.1 ± 12.9	44.6 ± 14.6	0.124
Sex—No. (%)			0.008
Female	14 (31.8)	4 (8.0)	
Male	30 (68.2)	46 (92.0)	
Ethnic group—No. (%)			0.191
White	40 (90.9)	50 (100.0)	
Black	1 (2.3)	0 (0.0)	
Mixed race	2 (4.5)	0 (0.0)	
Asian descent	1 (2.3)	0 (0.0)	
Educational level—No. (%)			0.753
Primary education	16 (36.4)	22 (44.0)	
Secondary education	19 (43.2)	19 (38.0)	
Higher education	9 (20.5)	9 (18.0)	
Current comorbidities or diseases (yes)—No. (%)	21 (47.7)	23 (46.0)	1.000
Hypertension	3 (6.8)	13 (26.0)	0.028
Diabetes mellitus	1 (2.3)	3 (6.0)	0.620
Dyslipidemia	0 (0.0)	6 (12.0)	0.028
Hepatitis	2 (4.5)	0 (0.0)	0.216
Common mental disorder	4 (9.1)	3 (6.0)	0.702
Musculoskeletal/rheumatologic disease	8 (18.2)	2 (4.0)	0.042
Other conditions	7 (15.9)	7 (14.0)	1.000
Use of medications—No. (%)	12 (27.3)	25 (50.0)	0.041
Use of herbal teas, supplements, and/or vitamins—No. (%)	8 (18.2)	1 (2.0)	0.011
Alcohol consumption—No. (%)	30 (68.2)	45 (90.0)	0.018
Smoking—No. (%)	0 (0.0)	2 (4.0)	0.497

**Table 2 ijerph-23-00034-t002:** Biochemical and genotoxicity biomarkers in the non-exposed and exposed groups.

Variables	Non-Exposed Group (*n* = 44)	Exposed Group (*n* = 50)	*p*-Value
ETU indicator—median (P25–P75)	0.05 (0.05–1.46)	168.9 (143.8–220.6)	<0.010
Binucleated—mean ± SD	28.2 ± 11.5	24.9 ± 10.1	0.139
Micronuclei—median (P25–P75)	8 (3.3–14.8)	9.5 (6–16)	0.105
BUD + BE—median (P25–P75)	2 (1–7)	4 (3–7)	0.094
Damage—median (P25–P75)	0 (0–20)	25 (0–41.5)	<0.001

Abbreviations: ETU: ethylene thiourea (metabolite of Mancozeb); BUD + BE: nuclear buds + broken eggs (DNA damage markers).

**Table 3 ijerph-23-00034-t003:** Multiple linear regression analysis of exposure effects, adjusted for confounding variables.

Variables	b (95% CI)	*p*-Value
AST *	7.90 (−0.30 to 16.1)	0.059
ALT **	8.02 (−4.06 to 20.1)	0.190
Creatinine ***	0.03 (−0.02 to 0.08)	0.219
Glucose *	−14.2 (−27.2 to −1.08)	0.034
Total cholesterol ***	−14.8 (−37.1 to 7.56)	0.192
LDL ***	−18.1 (−38.2 to 2.03)	0.077
ETU indicator *	213.8 (168.5 to 259.1)	<0.001
Oxidative stress protein *	−0.28 (−0.43 to −0.14)	<0.001
TBARS *	0.52 (0.46 to 0.57)	<0.001
Catalase *	6.38 (5.63 to 7.13)	<0.001
SOD *	−496.6 (−540.4 to −452.8)	<0.001
DNA damage	19.6 (8.13 to 31.0)	0.001

Adjustment notes: * Adjusted for hypertension, dyslipidemia, alcohol consumption, musculoskeletal disease, and elevated BMI. ** Adjusted for hypertension, dyslipidemia, alcohol consumption, and elevated BMI. *** Adjusted for hypertension, dyslipidemia, alcohol consumption, elevated BMI, age, and sex.

## Data Availability

The data presented in this study is available upon request from the corresponding author. This study focused exclusively on exposure to the fungicide Mancozeb, excluding the assessment of other groups of pesticides, such as organophosphates, carbamates, or quaternary compounds, which represents a recognized limitation.

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
