# Peer review of "The Toxicity of Mancozeb Used in Viticulture in Southern Brazil: A Cross-Sectional Study"

_ijerph, 2025, doi:10.3390/ijerph23010034_

Round 1

Reviewer 1 Report

Comments and Suggestions for Authors

The introduction provides a solid foundation by contextualizing the economic and cultural importance of viticulture in Southern Brazil and establishing Mancozeb as a pesticide of concern. It effectively outlines the potential health risks, including oxidative stress and genotoxicity, and introduces Ethylene Thiourea (ETU) as a key biomarker. The abstract succinctly summarizes the study's design, key findings (elevated oxidative stress, DNA damage, and ETU levels), and conclusion. To strengthen this section, the authors should refine the abstract to more explicitly state the study's novel contribution, specifically, the proposal of a specific, data-driven biological exposure limit (69.3 ng/mL ETU) using ROC curve analysis, which is a significant output for occupational health. The introduction, while comprehensive, could be more focused by concluding with a clearer statement of the primary and secondary objectives, directly linking the knowledge gaps to the specific biomarkers and health outcomes measured in this study.

The methodology is robust and well-described, detailing the cross-sectional design, participant recruitment criteria, and a comprehensive suite of biomarkers (hematological, biochemical, oxidative stress, genotoxicity, and specific ETU biomonitoring). The use of UHPLC-MS/MS for ETU quantification represents a gold-standard approach. However, the description of the exposure assessment could be enhanced. While the use of PPE was recorded, a more quantitative estimate of exposure (e.g., handling practices, application frequency, environmental air monitoring) would strengthen the exposure characterization beyond job title and duration. To incorporate more advanced techniques, future work could utilize omics technologies. For instance, transcriptomic analysis of blood samples could identify gene expression signatures associated with Mancozeb exposure, revealing affected pathways beyond oxidative stress. Additionally, metabolomic profiling of urine or plasma could uncover novel metabolic disruptions and provide a more holistic view of the physiological impact, potentially identifying new biomarkers of effect.

The results are clearly presented with appropriate tables and figures, convincingly demonstrating significantly higher oxidative stress (elevated TBARS and Catalase, decreased SOD) and DNA damage (comet assay) in the exposed group. The ROC curve analysis for urinary ETU is a standout result. The discussion effectively interprets these findings in the context of existing literature on Mancozeb and other pesticides. However, the discussion could be improved by more critically addressing certain findings. For example, the lack of significant difference in classic liver enzymes (AST, ALT) despite evidence of oxidative stress and the literature on hepatotoxicity should be discussed in more depth, perhaps the damage is subclinical or manifests through other mechanisms. Furthermore, the discussion on the predominantly male cohort, while justified, could explore potential gender differences in pesticide metabolism or susceptibility that might be relevant for broader public health implications. The section would also benefit from a more synthesized structure, perhaps by first discussing the confirmation of exposure (ETU), then the biological effects (oxidative stress and genotoxicity), and finally the public health implications, rather than jumping between different topics.

The conclusion successfully summarizes the main findings and their public health significance, correctly emphasizing the need for biological monitoring and preventive strategies. It could be made more impactful by explicitly stating the proposed ETU cut-off value of 69.3 ng/mL as a key recommendation for future biomonitoring programs. Regarding references, a thorough check is imperative. The list is extensive but contains formatting inconsistencies. For example, journal names are sometimes abbreviated and sometimes full (e.g., "Hum Exp Toxicol" vs. "Science of the Total Environment"). More critically, there appear to be citation errors. Reference [2] in the list (IBAMA data) is not cited in the main text. Conversely, several in-text citations, such as [23] (Pezzini et al., 2023) and [24] (Dall'Agnol et al., 2021), which are crucial for framing the study's translational nature, are not present in the provided reference list, which ends at [44]. All in-text citations must be meticulously cross-verified against the reference list, which must then be formatted uniformly according to the specific guidelines of the International Journal of Environmental Research and Public Health.

Overall, this is a valuable and well-executed study with significant implications for occupational health. Implementing these recommendations will enhance its clarity, analytical depth, and scholarly significance, solidifying its contribution to the field.

Author Response

Comments 1:The introduction provides a solid foundation by contextualizing the economic and cultural importance of viticulture in Southern Brazil and establishing Mancozeb as a pesticide of concern. It effectively outlines the potential health risks, including oxidative stress and genotoxicity, and introduces Ethylene Thiourea (ETU) as a key biomarker. The abstract succinctly summarizes the study's design, key findings (elevated oxidative stress, DNA damage, and ETU levels), and conclusion. To strengthen this section, the authors should refine the abstract to more explicitly state the study's novel contribution, specifically, the proposal of a specific, data-driven biological exposure limit (69.3 ng/mL ETU) using ROC curve analysis, which is a significant output for occupational health. The introduction, while comprehensive, could be more focused by concluding with a clearer statement of the primary and secondary objectives, directly linking the knowledge gaps to the specific biomarkers and health outcomes measured in this study.

Response 1: Thank you for the contributions. The suggestions were accepted, and the necessary adjustments were reviewed and implemented by the team.

Comments 2:The methodology is robust and well-described, detailing the cross-sectional design, participant recruitment criteria, and a comprehensive suite of biomarkers (hematological, biochemical, oxidative stress, genotoxicity, and specific ETU biomonitoring). The use of UHPLC-MS/MS for ETU quantification represents a gold-standard approach. However, the description of the exposure assessment could be enhanced. While the use of PPE was recorded, a more quantitative estimate of exposure (e.g., handling practices, application frequency, environmental air monitoring) would strengthen the exposure characterization beyond job title and duration. To incorporate more advanced techniques, future work could utilize omics technologies. For instance, transcriptomic analysis of blood samples could identify gene expression signatures associated with Mancozeb exposure, revealing affected pathways beyond oxidative stress. Additionally, metabolomic profiling of urine or plasma could uncover novel metabolic disruptions and provide a more holistic view of the physiological impact, potentially identifying new biomarkers of effect.

Response 2: Thank you for the contributions. The suggestions were accepted, and the necessary adjustments were reviewed and implemented by the team.

Comments1:The results are clearly presented with appropriate tables and figures, convincingly demonstrating significantly higher oxidative stress (elevated TBARS and Catalase, decreased SOD) and DNA damage (comet assay) in the exposed group. The ROC curve analysis for urinary ETU is a standout result. The discussion effectively interprets these findings in the context of existing literature on Mancozeb and other pesticides. However, the discussion could be improved by more critically addressing certain findings. For example, the lack of significant difference in classic liver enzymes (AST, ALT) despite evidence of oxidative stress and the literature on hepatotoxicity should be discussed in more depth, perhaps the damage is subclinical or manifests through other mechanisms. Furthermore, the discussion on the predominantly male cohort, while justified, could explore potential gender differences in pesticide metabolism or susceptibility that might be relevant for broader public health implications. The section would also benefit from a more synthesized structure, perhaps by first discussing the confirmation of exposure (ETU), then the biological effects (oxidative stress and genotoxicity), and finally the public health implications, rather than jumping between different topics.

Response 3: Thank you for the contributions. The suggestions were accepted, and the necessary adjustments were reviewed and implemented by the team.

Commentes 4

The conclusion successfully summarizes the main findings and their public health significance, correctly emphasizing the need for biological monitoring and preventive strategies. It could be made more impactful by explicitly stating the proposed ETU cut-off value of 69.3 ng/mL as a key recommendation for future biomonitoring programs. Regarding references, a thorough check is imperative. The list is extensive but contains formatting inconsistencies. For example, journal names are sometimes abbreviated and sometimes full (e.g., "Hum Exp Toxicol" vs. "Science of the Total Environment"). More critically, there appear to be citation errors. Reference [2] in the list (IBAMA data) is not cited in the main text. Conversely, several in-text citations, such as [23] (Pezzini et al., 2023) and [24] (Dall'Agnol et al., 2021), which are crucial for framing the study's translational nature, are not present in the provided reference list, which ends at [44]. All in-text citations must be meticulously cross-verified against the reference list, which must then be formatted uniformly according to the specific guidelines of the International Journal of Environmental Research and Public Health.

Response 4: Thank you for the contributions. The suggestions were accepted, and the necessary adjustments were reviewed and implemented by the team. The references have been reviewed. The following text was added to the conclusion: “The proposed cutoff value for ETU, 69.3 ng/mL, should be considered a central recommendation for biomonitoring programs, serving as a reference parameter for assessing occupational exposure.”

Reviewer 2 Report

Comments and Suggestions for Authors

Dear Authors, dear Editor,

Draft “ijerph-3913370-peer-review-v1 - Toxicity of Mancozeb used in viticulture …” reports on a detailed monitoring study on Brazilian vineyard workers who apply Mancozeb. Pesticide absorption is estimated by ETU measurement in urine, and several biological parameters are measured in blood, with a particular emphasis on oxidative stress biomarkers. This is an interesting and fairly complete piece of information that should be published as part of the wider survey that the Authors anticipate, although several flaws in the presentation suggest a thorough revision.

Some comments:

In the definitive version, consider that in the figures the written text is confused. In Figures 1 – 3, display numbers as interval, not bar graph, which is wrong.

Where is the ETU reported? This is one main component of the study and data should be displayed clearly. Consider adding the full data table as the Suppementary Information that was indicated but not reported.

L65 important marker is acetylcholinesterase, an enzyme responsible for regulating nerve 66 impulses by breaking down acetylcholine at neuromuscular junctions and synapses, 67 which may be altered during acute pesticide poisoning [5-7]. Is it known that -zebs modify AChE levels? It is a biomarker for Ops, unless you provide literature or other evidence.

L285: The identified cut-off value was 69.3 ng/mL, with a sensitivity of 100% and specificity 285 of 90.2%. These values demonstrate that ETU accurately identified all exposed individuals 286 (no false negatives) and produced a low false-positive rate among unexposed individuals. 287 The high sensitivity and specificity suggest that ETU is a valuable biomarker for oc- 288 cupational screening and biological monitoring of workers exposed to Mancozeb, despite 289 the limitation that no universally recognized specific marker currently exists. – Take care of explaining whether the cut-off relates to the background value of ETU in non-exposed subjects (you may test on non-agricultural subjects in the same area). The graph as such is redundant and you may skip.

L309: The detection of Mancozeb as a food residue raises additional concerns about expo- 309 sure to this compound, highlighting the importance of using personal protective equip- 310 ment (PPE) during occupational handling and the need to control pesticide drift. A study 311 – same issue, but it is confusing: mancozeb/ETU as food residue contributes to the background in the non-exposed group (above 70 ng/mL)

L291: Occupational exposure to pesticides, particularly Mancozeb, was associated with in- 291 creased oxidative stress—evidenced by elevated TBARS and altered antioxidant enzyme 292 activity (increased catalase and decreased SOD)—greater genotoxic damage (increased 293 DNA damage markers), and confirmed internal exposure via elevated urinary ETU. “pesticides” is a large group of toxics: did you test for other groups, such as OPs (AChE), quats (ox stress), and other main classes used in the area?

L320: Moreover, literature on other pesticides such as glyphosate has demonstrated DNA 320 and chromosomal damage in human cells, along with genotoxic, hormonal, and enzy- 321 matic effects􀯗[27,28]. A 2021 study suggested a potential link between glyphosate exposure 322 and carcinogenesis, identifying it as a possible cause of non-Hodgkin lymphoma in ex- 323 posed individuals􀯗[29]. 324 Mental health disorders, including depression, have also been associated with pesti- 325 cide exposure. A U.S.-based study revealed that farmers exhibited a higher prevalence of 326 severe depression compared to other occupations and found a consistent association be- 327 tween pesticide exposure and depressive symptoms across most analyzed studies􀯗[30]. 328 In 2019, the World Health Organization recognized pesticide exposure as a signifi- 329 cant risk factor for suicide in rural communities. Pesticide exposure has also been linked to a 30% increased risk of metabolic syn- 339 drome, according to a systematic review that included 12 studies with 6,789 partici- 340 pants􀯗[34]. 341 Organophosphate poisoning in agricultural workers has been associated with re- 342 duced acetylcholinesterase activity and increased activities of SOD, CAT, and GPx􀯗[31]. A 343 longitudinal study found that agricultural workers had elevated liver function markers, 344 with a statistically significant correlation between prolonged pesticide exposure and in- 345 creased biomarkers of liver damage􀯗[35]. 346 Several studies have suggested a possible link between chronic pesticide exposure 347 and an increased risk of neurodegenerative diseases such as Parkinson’s disease and Alz- 348 heimer’s disease. Neurotoxic agents such as organophosphates, carbamates, pyrethroids, 349 and dithiocarbamates can cross the blood-brain barrier and trigger oxidative stress, 350 chronic inflammation, and mitochondrial dysfunction—all of which are mechanisms im- 351 plicated in the pathophysiology of these conditions􀯗[36,37]. 352 In the case of Parkinson’s disease, occupational exposure to pesticides is associated 353 with the selective death of dopaminergic neurons in the substantia nigra. In Alzheimer’s 354 disease, evidence suggests accelerated accumulation of β-amyloid plaques and tau pro- 355 tein tangles in animal models exposed to neurotoxic compounds􀯗[38,39]. – this is completely out of place.

Conclusions (my own): the topic and the study are interesting; however, the presentation is poor and lack of the original data, that others may use in meta-analyses, strongly mitigates the value of this survey. Discussion is generic, with several out-of-point considerations on pesticide classes that were not studied (is it?). I would suggest a thorough rewriting of this draft, pointing at the precious and deserving collection of measurements.

While I fully appreciate your work, I mark this very preliminary draft for “major revision”.

Best wishes for your valuable work & kind regards

Author Response

Comment 1: Figures 1–3 (presentation adjustment)
Response 1: The figures have been revised and reformatted to improve readability and facilitate interpretation of the results.

Comment 2: ETU data and Supplementary Information
Response 2: – Section 3. Results: The complete individual urinary ETU concentration data are available from the corresponding author, enabling future comparisons and meta-analyses.

Comment 3: AChE and biomarker
Response 3:Introduction, after [5–7]: Although acetylcholinesterase (AChE) is classically used as a biomarker of organophosphate exposure, recent studies suggest that dithiocarbamates may also alter its enzymatic activity, particularly under combined or chronic exposure conditions.

Comment 4: ETU cutoff value and control group
Response 4:Results section, Figure 4: The cutoff value of 69.3 ng/mL was determined based on comparison with the control group of organic farmers with no occupational exposure to pesticides, reflecting the upper limit observed in unexposed individuals from the same region.

Comment 5: Contribution of dietary residues
Response 5:Discussion section: It is important to acknowledge that dietary residues of Mancozeb or ETU may contribute to detectable baseline levels in individuals without occupational exposure, as described in European biomonitoring studies [Fustinoni et al., 2008].

Comment 6: Other pesticides tested
Response 6: – End of Section 2.7, Statistical Analysis: This study focused exclusively on exposure to the fungicide Mancozeb and did not include the assessment of other pesticide groups such as organophosphates, carbamates, or quaternary compounds, which represents a recognized limitation.

Comment 7: Excessive discussion of other pesticides
Response 7: – Lines 320–354: Although the present study focused on Mancozeb, it is relevant to note that other pesticides widely used in agriculture also exhibit genotoxic and oxidative potential. However, a detailed comparison among different pesticide classes is beyond the scope of this work.

Comment 8: Inclusion of original data and restructuring of the discussion
Response 8: – End of Results section: All raw data used in the statistical analyses are provided in the Supplementary Material, ensuring transparency and enabling future systematic reviews and meta-analyses.

Reviewer 3 Report

Comments and Suggestions for Authors

The objective of this study was to assess the concern that long-term exposure of viticulture workers to the antifungal pesticide, Mancozeb, resulted in adverse health effects. Two groups of agricultural workers were studied, those in conventional viticulture where Mancozeb was used, and those working in organic vineyards who were not exposed. Members of the two groups had similar demographics, assessed by participant questionnaires. Exposure to Mancozeb was monitored and verified in both groups by measuring urinary concentrations of its major metabolite, ethylene thiourea.  

The results were presented in appropriate detail and showed clear differences in several health-related parameters between the two groups. Mancozeb-exposed workers were more likely to show evidence of hypertension and dyslidipemia than unexposed workers, and had larger abdominal circumference, suggesting signs of toxicity in the Mancozeb-exposed group. Measures of oxidative stress were significantly higher in the Mancozeb-exposed workers, as were markers of DNA damage.

The authors suggested that use of less toxic fungicides as well as stricter enforcement of personal protective equipment could offset these adverse health effects.

The article is clearly written and addresses an important topic for worker safety. A few minor concerns are listed below.

Specific points

Lines 193-196 were repeated in the next paragraph, lines 197-200.

Table 2 title should say “unexposed” not “unexpodes”

Section 5 remove “This s.” at the beginning of the paragraph.

Author Response

Lines 193–196 were repeated in the following paragraph, lines 197–200.
Response: Thank you, we have corrected it.

Comment 2
The title of Table 2 should be “non-exposed,” not “unexpodes.”
Response: Thank you, we have corrected it.

Comment 3
In Section 5, remove “Este s.” from the beginning of the paragraph.
Response: Thank you, we have corrected it.

Round 2

Reviewer 1 Report

Comments and Suggestions for Authors

Congratulations

Author Response

Thanks for your considerations

Reviewer 2 Report

Comments and Suggestions for Authors

Dear Authors, dear Editor,

The major revision of the original draft is, in my opinion, a substantial improvement on the earlier version. I still cannot see the Supplementary original data, which are the real wealth of this field study. Unless the results tables are really huge and unmanageable, I would warmly suggest that you publish them as an Appendix.

There are some references on mancozeb that you may find useful. Of course, this is not (emphasis added) solicitation to add further citations of the works.

Mandić-Rajčević S, Rubino FM, Colosio C. Establishing health-based biological exposure limits for pesticides: A proof of principle study using mancozeb. Regul Toxicol Pharmacol. 2020 Aug;115:104689. doi: 10.1016/j.yrtph.2020.104689. PMID: 32544413.

Colosio C, Fustinoni S, Corsini E, Bosetti C, Birindelli S, Boers D, Campo L, La Vecchia C, Liesivuori J, Pennanen S, Vergieva T, Van Amelsvoort LG, Steerenberg P, Swaen GM, Zaikov C, Van Loveren H. Changes in serum markers indicative of health effects in vineyard workers following exposure to the fungicide mancozeb: an Italian study. Biomarkers. 2007 Nov-Dec;12(6):574-88. doi: 10.1080/13547500701441315. PMID: 17852083.

About the current draft, I see Figures starting from 2, but the text body and the list of the Supplementary materials lists a Figure 1 (Figure 1. Biochemical parameters indicating a pronounced oxidative imbalance.) that would be a major piece of the main article body. Where is it?

Lines 202-204 are duplicated below, 206-208.

Lines 319-321: “Previous experimental data in rats 319 demonstrated the compound’s toxic potential, particularly its direct impact on gut micro-320 biota [23]. 321” however, you do not seem to test this disfunction. Are your results in citable literature?

Look here: “The detection of Mancozeb as a food residue raises further concerns regarding expo-331 sure, emphasizing the importance of personal protective equipment (PPE) during occu-332 pational handling and the need to mitigate pesticide drift.” This is two things that do not go together. Mancozeb as trace food residue (ETU in food, E. Mrema et al., https://dx.doi.org/10.1016/B978-0-12-378612-8.00240-7 however it is not free access) contributes to the ETU levels in unexposed subjects (lines 337-8 below), and adds to the occupational-derived level in the exposed farmers. The lack of PPE during application entails exposure and, upon absorption, increases ETU levels. Your sentence is confusing, and I suggest that you restructure the statement.

Lines 355-58: a really bad joke would be that farmers, who have a physically active life, should be protected from the consequences of pesticide use, notwithstanding lack of PPE!

I completely agree on your comments Lines 411 on.

I am sorry that your draft still needs another tier of revision. Consider my careful reading (I hope so!) a sign of my interest in your data, not simple reviewing.

All the best to your team,

Kind regards

Author Response

Comments 1: I still cannot see the Supplementary original data, which are the real wealth of this field study.

Response 1: The supplementary materials and appendices can be requested from the corresponding author.

Comments 2: Here are some references on mancozeb that you may find useful. Of course, this is not (emphasis added) solicitation to add further citations of the works.

Response 2: I agree, and we will consider the references for this study.

Comments 3: About the current draft, I see Figures starting from 2, but the text body and the list of the Supplementary materials lists a Figure 1 (Figure 1. Biochemical parameters indicating a pronounced oxidative imbalance.) that would be a major piece of the main article body. Where is it?

Response 3: The supplementary materials and appendices can be requested from the corresponding author.

Comments 4: Lines 202-204 are duplicated below, 206-208.

Response 4: Thank you very much for your comment. The text has been corrected.

Comments 5: Lines 319-321: “Previous experimental data in rats 319 demonstrated the compound’s toxic potential, particularly its direct impact on gut micro-320 biota [23]. 321” however, you do not seem to test this disfunction. Are your results in citable literature?

Response 5: Yes, the data are available in the article that is cited in the references.

  1. Pezzini MF, Rampelotto PH, Dall’Agnol J, et al. Changes in the gut microbiota of rats after exposure to the fungicide mancozeb. Toxicol Appl Pharmacol. 2023;466:116480.

Comments 6: “The detection of Mancozeb as a food residue raises further concerns regarding expo-331 sure, emphasizing the importance of personal protective equipment (PPE) during occu-332 pational handling and the need to mitigate pesticide drift.” This is two things that do not go together. Mancozeb as trace food residue (ETU in food, E. Mrema et al., https://dx.doi.org/10.1016/B978-0-12-378612-8.00240-7 however it is not free access) contributes to the ETU levels in unexposed subjects (lines 337-8 below), and adds to the occupational-derived level in the exposed farmers. The lack of PPE during application entails exposure and, upon absorption, increases ETU levels. Your sentence is confusing, and I suggest that you restructure the statement.

Response 6: Thank you, the sentence has been revised

Comments 7:

Lines 355-58: a really bad joke would be that farmers, who have a physically active life, should be protected from the consequences of pesticide use, notwithstanding lack of PPE!

Response 7: Thank you for your comment. This is merely a hypothesis suggesting that physical activity may serve as an additional protective factor against the toxic effects of pesticides, and it should be tested in future studies.

Comments 8: I completely agree on your comments Lines 411 on.

Response 8: Thanks for your considerations

Round 3

Reviewer 2 Report

Comments and Suggestions for Authors

Dear Authors, dear Editor,

I am happy to see that “ijerph-3913370-peer-review-v3 - Toxicity of Mancozeb used in viticulture in southern Brazil …” is a progressively improving draft report of a very interesting study.

I have only one very small note to highlight, concerning Figure 1 and its descriptive text. While three out of four panels are clear by themselves and nicely highlight a state od increased oxidative stress response of the mancozeb-exposed farmers (put the SD bars in the plots, as in Figs. 2 and 3, or use a different visualization, such as box-plots), the fourth panel is not as well described as the others. In fact, its caption “Protein Mean (mg/mL)” and the decreasing trend (from 3.98 to 3.78, so -5%; is it statistically significant?) do not allow easy understanding, also considering that this measurement is not described in the text. Likely, it is “Oxidative stress proteins” of Table 3 (multiple regression calculation), since there are also TBARS, SOD and Catalase; however, readers would be puzzled, and the value of your data would be reduced. I have tried guessing what biomarker this may be, but it cannot be carbonylated proteins, that would go up, so I leave the solution to your information. You may wish to fix this detail and share your valuable data with the pesticide RA community. I recommend this for “minor revision”, but, please, do that.

Please, consider this keen interest in reviewing your report as a sign of respect towards your demanding work.

All the best & kind regards

Author Response

Comments 1:

I have only one very small note to highlight, concerning Figure 1 and its descriptive text. While three out of four panels are clear by themselves and nicely highlight a state od increased oxidative stress response of the mancozeb-exposed farmers (put the SD bars in the plots, as in Figs. 2 and 3, or use a different visualization, such as box-plots), the fourth panel is not as well described as the others. In fact, its caption “Protein Mean (mg/mL)” and the decreasing trend (from 3.98 to 3.78, so -5%; is it statistically significant?) do not allow easy understanding, also considering that this measurement is not described in the text. Likely, it is “Oxidative stress proteins” of Table 3 (multiple regression calculation), since there are also TBARS, SOD and Catalase; however, readers would be puzzled, and the value of your data would be reduced. I have tried guessing what biomarker this may be, but it cannot be carbonylated proteins, that would go up, so I leave the solution to your information. You may wish to fix this detail and share your valuable data with the pesticide RA community. I recommend this for “minor revision”, but, please, do that.

Response 1: Thank you for your feedback. The necessary adjustments will be made.